# Integrin Conformational Dynamics and Mechanotransduction

**DOI:** 10.3390/cells11223584

**Published:** 2022-11-12

**Authors:** Reza Kolasangiani, Tamara C. Bidone, Martin A. Schwartz

**Affiliations:** 1Department of Biomedical Engineering, University of Utah, Salt Lake City, UT 84112, USA; 2Scientific Computing and Imaging Institute, University of Utah, Salt Lake City, UT 84112, USA; 3Yale Cardiovascular Research Center, Department of Cardiovascular Medicine, Yale University, New Haven, CT 06520, USA; 4Department of Cell Biology, Yale University, New Haven, CT 06520, USA; 5Department of Biomedical Engineering, School of Engineering and Applied Science, Yale University, New Haven, CT 06520, USA

**Keywords:** integrin, mechanotransduction, conformational activation

## Abstract

The function of the integrin family of receptors as central mediators of cell-extracellular matrix (ECM) and cell–cell adhesion requires a remarkable convergence of interactions and influences. Integrins must be anchored to the cytoskeleton and bound to extracellular ligands in order to provide firm adhesion, with force transmission across this linkage conferring tissue integrity. Integrin affinity to ligands is highly regulated by cell signaling pathways, altering affinity constants by 1000-fold or more, via a series of long-range conformational transitions. In this review, we first summarize basic, well-known features of integrin conformational states and then focus on new information concerning the impact of mechanical forces on these states and interstate transitions. We also discuss how these effects may impact mechansensitive cell functions and identify unanswered questions for future studies.

## 1. Introduction

Over three decades after receiving its name, the integrin family of transmembrane adhesion receptors is now recognized as the core component of the most studied cellular mechanotransduction pathways to date [1,2,3,4,5]. Mechanotransduction is the process by which a cell converts mechanical stimuli into biochemical signals that regulate cell activity. It occurs in a vast range of physiological processes, including embryonic morphogenesis [6], wound healing [7], and tissue regeneration [8], and in pathologies such as cancer [9,10], fibrosis of many organs [11,12], and cardiovascular diseases [13]. Well-studied, physiologically important instances include the functional adaptation of osteoblasts and osteoclasts to mechanical loading during the formation, repair, and remodeling of bone tissue, and the response of endothelial cells to fluid shear stress that determines local susceptibility to atherosclerosis. Integrins have been implicated in most of these processes, both physiological and pathological.

Integrins are noncovalent heterodimers formed by an α and a β subunit. Both subunits consist of an extracellular globular ‘head’ with a rod-like ‘leg’, a membrane-spanning helix, and a short cytoplasmic tail (Figure 1A). The main cytoskeletal anchorage of integrin is through the β cytoplasmic tail, which binds accessory proteins such as talin, integrin-linked kinase, and filamin, to connect to actin filaments. There are 18 α-subunits and 8 β-subunits, forming 24 distinct αβ integrin combinations [14]. The combination of α- and β-subunits defines its cell type specificity and the affinity for different ligands, such as collagen, fibronectin, and laminin [15,16]. Integrins were named to indicate their roles as integral membrane proteins that physically connect the cell cytoskeleton to the extracellular matrix (ECM), thus integrating internal and external structures [17]. Subsequent work demonstrated that integrins are not only structural but also signaling receptors (reviewed in [18,19]), thus introducing the possibility that force across the integrins might also regulate signaling pathways. The growth of mechanobiology as a discipline has thus fueled growing interest in understanding integrin not only as a structural macromolecule, but also as a mechanotransducer that is able to sense forces, stresses, and deformations to elicit adaptive cell responses.

Mechanotransduction requires that integrins bear mechanical load. These are transmitted via noncovalent binding to extracellular ligands and cytoplasmic adapters that link the extracellular matrix and the contractile actin cytoskeleton. Force can originate either from intracellular contractile acto-myosin or from extracellular deformations but with similar consequences. Integrins respond to forces through changes in adhesion assembly, and in the organization and function of the entire structure, from extracellular matrix fibrils that require tension to assemble [20], to the intracellular proteins that associate with integrin and link to actin filaments [21,22], to the contractile acto-myosin bundles that generate force [23]. All of these are important in cellular responses to force; however, in this review we focus on the integrins themselves and their conformational dynamics.

**Figure 1 cells-11-03584-f001:**
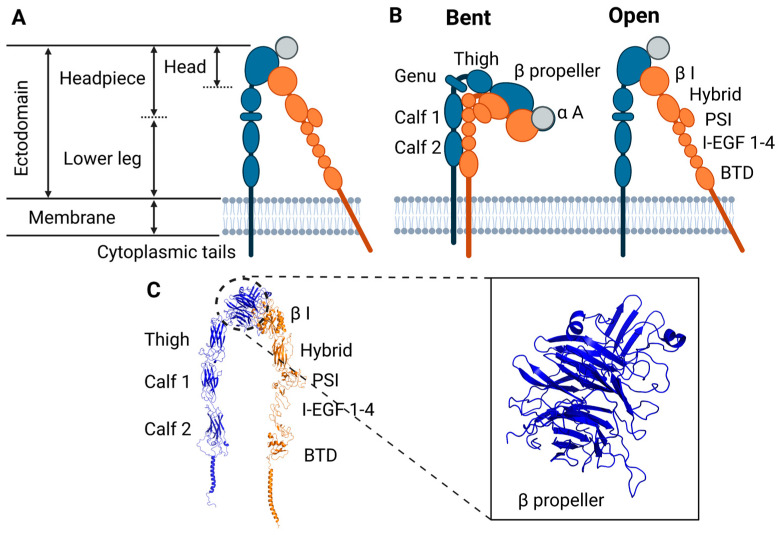
Schematics of integrin in bent and extended conformations. (**A**). Integrins are non-covalent heterodimers comprised of an α subunit (blue) and a β subunit (orange). Both subunits include an extracellular region, or ectodomain, which includes the headpiece and legs, a transmembrane helix, and a cytoplasmic domain. (**B**). Bent and extended conformations of integrin. The α subunit (blue) consists of an N-terminal, a β-propeller, a thigh domain, two calf domains, transmembrane α helix, and cytoplasmic α tail. The α A domain is represented in gray. The β subunit (orange) consists of an N-terminal β-I domain followed by the hybrid, the plexin-semaphorin-integrin domain (PSI), four cysteine-rich epidermal growth factor (EGF) modules (I-EGF) 1-4, β-Tail domain (BTD), transmembrane β helix, and cytoplasmic β tail. (**C**). Ribbon representation of extended-open αIIbβ3 from cryo-EM reconstructions [24]. Secondary structure elements of the α (blue) and β (orange) subunit are shown. The α subunit seven-bladed β-propeller domain is represented in the zoomed section.

Integrins exist in different types of adhesions that can be classified as nascent adhesions, focal complexes, and mature focal adhesions [25]. During directional cell migration, the polymerization of actin filaments against the cell leading edge establishes a protrusion that transport integrins to the leading edge. As integrins bind the ECM, the recruitment of intracellular adaptors leads to the sequestration of more integrins, with the formation of small (10–100 nm in diameter), short lived (1–2 min) integrin clusters named nascent adhesions. Nascent adhesions either disassemble or stabilize and elongate into focal complexes (1–2 µm, lifetimes of a few minutes), with recruitment of myosin, which provides higher contractile forces [3,26]; recruitment of adapters, such as talin, vinculin, and α-actinin; and activation of tyrosine kinases [27,28,29,30,31,32,33,34]. A subset of focal complexes further mature into longer and more stable focal adhesions (several µm long, lifetimes of 10’s of minutes) that anchor large actin stress fibers to the ECM matrix [32,35]. In migrating cells, these events form a cycle, whereafter focal adhesion disassembly, integrins, and other components subsequently recycle to the cell’s leading edge to create new nascent adhesions [3,31,36,37].

During the assembly of adhesions, from nascent to focal complexes and mature focal adhesions, integrins undergo dynamic transitions from the bent (inactive) to the extended (active) conformations [3,36,38]. These transitions involve long-range motions of secondary structure elements and functional domains, including significant reorientation of the ligand binding interface, which increases affinity for ECM ligands (>1000-fold in some cases) [39,40,41], and the lifetime of the ligand-bound state (~5-fold) [42].

How these conformational transitions impact integrin functions as a structural linker, a mechanotransducer, and a signaling receptor is the subject of this review. We will focus on the current evidence supporting a role for integrin conformation in mechanotransduction by discussing how integrins change conformation under force and identifying prevailing hypotheses for future studies. We will start by focusing on how conformational changes of integrins are implicated in cell adhesiveness. Then, we will summarize the experimentally detected ranges of force on adhesion complexes and on single adhesion molecules and present the characteristics of integrin conformation and molecular rearrangements underlying its interconversion from inactive to active conformations. Lastly, we will identify important questions for future research needed to better understand integrin conformational dynamics in physiology and disease.

## 2. Integrin Conformation in Cell Adhesiveness

Changes in integrin conformation tightly regulate its affinity for ligand binding, thus determining cellular adhesion to ligands in the ECM or on other cells. Electron microscopy and X-ray crystallography studies using ECM ligands or activating and inhibitory antibodies have shown that integrins adopt a range of conformations [24,42,43,44,45,46,47,48,49,50]. For β_1_, β_2,_ and β_3_ integrins, three main conformations have been visualized: bent-closed, extended-closed, and extended-open [43,44,45,46,47,48,49,50,51,52]. For these integrins, high concentrations of ligands induce the extended-open conformation, which binds ligands with higher affinity than the closed conformations, as reported from several techniques including electron microscopy (EM) combined with cell-surface affinity measurements and conformation-specific Fabs [47,48,49,50,53,54,55].

Attachment and spreading of cells on ligand-coated substrates to assess cell adhesiveness indicate a role for mechanics in these processes. Cells adhere and spread less well on soft substrates, which impacts a range of signaling and gene expression pathways, a process termed stiffness sensing [56]. The differences in spread area of cells on stiff versus soft substrates correlate with changes in the morphology and turnover of focal adhesions. On compliant substrates, focal adhesions present irregular shapes and are highly dynamic, whereas those on more rigid substrates have regular shapes and are more stable [57].

Studies of cell spreading in different mechanical environments have revealed that integrin conformational activation depends on ECM stiffness and cytoskeletal forces [58,59]. Analysis of a collection of equivalently activated α_V_β_3_ mutants on elastic substrates of varying stiffness showed that these mutants shift cell traction and spreading towards lower stiffnesses [59]. Multiscale modeling based on atomistic simulations demonstrated that these mutants present different conformational flexibilities and shift the force required for integrin extension towards lower values, indicating a critical role for force-induced conformational deformations of integrin in stiffness sensing [59]. Similarly, activating exogenous integrins with manganese ions stabilized nascent adhesions and increases cell spreading on soft surfaces [58]. Collectively, these studies support a picture in which force impacts integrin conformational activation to regulate cell adhesiveness and spreading.

This concept is valid also in other types of cell adhesiveness measurements. For example, measures of platelet aggregation have implicated force-dependent integrin conformations that are intermediate between bent and extended conformations [42]. Using fluorescence dual biomembrane force probe, microfluidics, and cone-and-plate rheometry, α_IIB_β_3_ integrins under various mechanical stimuli were found in conformational intermediates with ectodomain extensions and affinities for binding ECM ligands that are intermediate between those of the bent and extended conformations [42]. The conformational intermediates presented lifetimes of ligand-bound states that are intermediate between those of inactive and active conformations [42]. Interestingly, these integrin force-dependent intermediates were sustainably expressed on live platelets and promoted platelet aggregation. The variations in the spread area of cells on different substrates and in the amount of cell aggregation from the different integrin conformations indicate a functional role of intermediates, in which conformational changes of integrins are correlated with gradual changes in the activity of the receptor.

Taken together, these studies on the relation between integrin conformation, adhesion dynamics, and cell adhesiveness revealed the existence of functionally important integrin conformations that are different from the well-known bent and extended conformations. These conformations are mechanosensitive and depend both structurally and functionally on force, further suggesting that they can underlie distinct signaling mechanisms.

## 3. Ranges of Mechanical Force on Integrins

Force on integrin adhesions can be measured in bulk, for the whole adhesion or whole cell, or per molecule. Bulk measurements use deformable substrates, such as polymer films or gels, or cantilevers to determine traction stresses [60,61,62,63,64,65]; per molecule measurements use fluorescent sensor modules containing molecular springs of defined properties [66,67]. Some per molecule measurements use substrates to which adhesive ligands are attached via double stranded DNA linkages that break under defined loads to report maximal rather than average force per molecule [68,69,70]. Average forces on single talin molecules that transmit force between integrins and actin filaments are in the 5 pN range, with considerable variation both within the same adhesions and in different adhesions within the same cell [67,71]. However, measurements of maximal force per integrin report higher levels, up to 40 pN [69,70]. These values may be somewhat biased by the high loading rate used for calibration; when exerted for longer times, bonds break at lower forces [37]. However, a tension sensor that reversibly unfolds at 7–11 pN also shows that a significant fraction of talin molecules reached this higher force level [72].

Traction force microscopy with smooth muscle cells and mouse fibroblasts combined with estimates of numbers of engaged integrins have also suggested that the average force per integrin within focal adhesions is in the single pN range [73,74]. These results are thus comparable to the average force measured with the talin tension sensor [67]. As talin is the main link between integrins and F-actin, it is a reasonable surrogate for force/integrin. Average forces per integrin are thus likely to be in the mid-single pN range but with a fraction at substantially higher levels.

Using elastic micropatterned substrates, stationary fibroblasts, and cardiac myocytes on elastic gels exert adhesion forces that positively correlate with the area of the adhesion, which varies between 2–5 μm^2^ and corresponds to forces between 1–25 nN [73]. The stresses, calculated as force per unit area of adhesion, average ~5 nN/μm^2^ [73]. Forces at cell edges are always directed inward, which correlates with the direction of adhesion elongation. Force-induced elongation is typically a reversible process such that reducing cytoskeletal force leads to adhesion shrinkage [73].

It is generally considered that forces at a few pN are sufficient to regulate protein conformation and thus local molecular interactions [56]. This sensitivity of integrin to force allows cells to readily distinguish ligands that are fixed on the ECM versus presented in soluble form. Moreover, the large distance change and energetics associated with integrin activation suggest that a constant force as low as 2 pN applied across a single integrin-ligand bond should be enough to induce a conformational change from the bent to the extended-open conformation [40]. 

Single molecule sensors have reported differences in the magnitude of force on adhesion molecules during the initial assembly, during the “sliding” towards the cell center, which involves disassembly at the rear and assembly at the front, and in mature adhesions [71]. Generally, high tension on focal adhesion molecules is associated with either actin alignment [71] or with adhesion assembly and enlargement [66]. Conversely, low tension is associated with disassembly [66,75]. These results, again, make it plausible that effects of force on integrin conformation contribute to the observed adhesion dynamics. However, in the absence of appropriate methods to assess integrin conformation in situ at high resolution, this remains an open question.

## 4. How Force Affects Integrin Kinetics

Force strengthens integrin-mediated adhesions by promoting their growth, maturation, and recruitment of new cytoskeletal proteins that help resist the applied force. There is extensive evidence that force-dependent activation and stabilization of the talin-vinculin connection contributes to force-induced strengthening and maturation of adhesions [37,56,76,77,78]. By contrast, the involvement of integrin conformation in these processes is attractive but less well documented. Elegant biophysical studies have showed that forces on integrins increase their ligand-bound lifetimes, termed catch bond behavior, although, of course, very high forces decrease bond lifetime [79]. Applying force to a single α_L_β_2_ integrin in living cells via its ligand ICAM-1 increases the lifetime of the ligand-bound state [80,81]. Under rapid or/and cyclic forces, the increase of binding affinity is even more efficient, with bond stabilization persisting well after the force is removed [82], indicating a form of molecular memory.

Experiments and simulations on adhesion proteins that show affinity shifts have revealed that integrin catch bond behavior is coupled to long range conformational rearrangements of the receptor far from the binding site [83,84,85]. Binding to ECM ligands is an essential prerequisite for force application on integrins, with the force itself deriving from some combination of internal actomyosin contractility and/or externally applied strain. Force can deform the binding site and propagate through the protein structure. Ample evidence currently suggests that integrin can bind ligands in bent or intermediate conformations [24,86,87,88]. For example, α_v_β_8_ integrins with closed legs can bind ligands [89]. Additionally, closed conformations of β_1_ integrins present on-rates for ligand binding up to ~20-fold higher than the extended-open conformation [90]. These results suggest that in some cases conformational extension occurs after ligand binding [91]. Taken together, these studies indicate that ligand binding may precede inside-out signaling and that the conformational transitions of integrin plausibly contribute to cell responses to forces.

In regard to ECM stiffness sensing, cells in or on soft ECM substrates not only spread less but also reduce their contractility and exert lower force on the matrix, compared to cells in or on stiff materials [92,93]. According to a molecular clutch mechanism, the critical variable in stiffness sensing is loading rate, such that on soft substrates, the deformation of the substrate buffers the applied load to decrease the rate [94,95]. This is an active mechanosensing process whereby cells exert force on the ECM and “measure” the resultant displacement or force building/loading rate, which determines the lifetime of bonds within the ECM-integrin-cytoskeleton linkage and thus determines adhesion stability and signaling outputs. Combining the focal adhesion clutch model with knowledge of integrin catch bond behavior thus strongly implies that force on integrins that stabilize the ligand-bound, cytoskeletal-anchored conformation contributes to focal adhesion growth and stability under tension.

## 5. Integrin Conformation and Conformational Dynamics

Integrin conformational transitions are thought to contribute to its interrelated but distinct functions of mechanotransducer, force transmitter, and signal transducer. Integrin α and β subunits consist of consecutive domains connected by loops and flexible linkers (Figure 1B). The extracellular region of the α subunit contains an N-terminal domain followed by a large seven-bladed β-propeller domain, followed by the thigh domain, and two calf domains (Figure 1B,C). Nine of 18 *α* subunits have an α-A domain (I or inserted domain), consisting of approximately 200 amino acids, located between blades 2 and 3 of the β-propeller (gray domain in Figure 1B) [96]. The α-I domain assumes a Rossman fold with five β-sheets surrounded by seven α helices, similar to von Willebrand A domains [97,98]. The β subunit extracellular region consists of an N-terminal β-I domain (that resembles the *α*-I domain) followed by the hybrid domain, the plexin-semaphorin-integrin domain (PSI), and four cysteine-rich epidermal growth factor (EGF) modules (I-EGF) 1-4 and β-T domains (Figure 1B). Both α and β subunits have a single membrane spanning helix that ends with a C-terminal cytoplasmic tail. While β cytoplasmic domains show substantial sequence homology consistent with conserved binding to talin and other adapters, *α* cytoplasmic domains show little homology. The transmembrane domains (TMD) of the α and *β* subunits (α-TMD and *β*-TMD, respectively) associate with each other, which helps stabilize the closed, low affinity conformation. During conformational activation from bent to extended conformations, the domains reorient and change relative positions (Figure 1B), including separation of the transmembrane and cytoplasmic domains that is critical in relaying long-range conformational changes to the extracellular domains [99,100]. Thus, mutations that disrupt cytoplasmic domain noncovalent interaction can activate integrins [101,102,103,104].

Integrin α_IIB_β_3_ has been resolved in bent, intermediate, and extended conformations (Figure 2), in which the intermediate conformations present closed subunits and varying degrees of flattening between the headpiece and the legs, resulting in different degrees of vertical extension [24,105]. In the bent conformation, the α and β chains are close together, with the headpiece bent against the lower legs and two extremely bent ‘knees’ at the α genu and between I-EGF 1 and I-EGF 2 domains of the β subunit [47,48]. The ligand binding interface is situated between the α subunit β-propeller domain and the β subunit β-I domain, which is responsible for coordinating interactions with charged residues in ligands [98]. In bent integrin, this interface is close to and oriented towards the plasma membrane (Figure 2A). Bent integrin is ~10-11 nm high on the extracellular side. Although the molecular interfaces buried in the bent conformation of integrin are extensive, they have low shape complementarity and are hydrophilic and readily replaced by water molecules, so that destabilization of the bent conformation promotes the open conformation [47,48]. Upon extension, the headpiece can remain closed (Figure 2B), or can transition into an open conformation (Figure 2C), with the ligand binding interface oriented away, rather than towards the plasma membrane. When extended, the ligand binding affinity of integrin is high for large biological ligands, such as fibronectin and fibrinogen [47]. In the open, fully extended conformation, the head is separated from the legs and the α and β legs are apart (Figure 2C). In the extended-open conformation, the extracellular domains span a total vertical distance of about 15–20 nm from the membrane, for a total length of integrin up of ~28 nm from the cytoplasmic tails to the distal ligand binding interface [44,106,107].

Before ligand binding, three metal ion binding sites in the β subunit β-I domain are typically occupied by Ca^2+^ and Mg^2+^ [108]. The Mg^2+^ bound to the central, metal ion-dependent adhesion site (MIDAS) directly coordinates with the acidic sidechain shared by all integrin ligands [109]. The site adjacent to MIDAS (ADMIDAS) binds the Mn^2+^ ion that triggers activation of low-affinity integrins [110]. By competing with Ca^2+^ at the ADMIDAS, Mn^2+^ shifts integrin conformational equilibrium towards the open conformation [111]. On the other side of the MIDAS (Figure 3), the synergistic metal ion binding site (SyMBS), also termed ligand-associated metal ion binding site (LIMBS), binds the Mn^2+^. SyMBS was found bound to the ion simultaneously with the AMIDAS in the ligated crystal structure of α_V_β_3_ integrin. On the lower side of the β–propeller blades facing away from the ligand-binding surface, the last three or four blades of the β -propeller domain in the α subunit contain EF-hand domains that also bind Ca^2+^, to affect ligand binding allosterically [110,112].

In the ligand binding interface of integrin, the forces between ions and the chains of the ligand can drive changes in the positions of the aminoacids of the β-I and β-propeller domains of integrin, resulting in increased ligand binding affinity. Conversely, the changes in positions of these aminoacids upon integrin activation change the distances between ions and ligand chains, thus increasing electrostatic interaction potentials and integrin affinity for ligand binding. Therefore, ligand binding affects integrin conformation and integrin conformation affects ligand binding, creating a reciprocal relationship between structure and ligand-bound state of integrin.

## 6. The Effects of Force on Integrin Conformation

Several studies have reported that force promotes integrin conformational activation by governing the transition from bent to extended conformations [44,50,108,113,114,115,116]. The main structural modifications that integrins undergo during conformational activation have been identified; however, the exact mechanisms by which force promotes the changes in integrin conformation are poorly understood.

The two main conformational rearrangements during integrin activation are hinging of the legs in the transition from bent to extended and swing out of the β subunit hybrid domain. SMD analysis and single-molecule experiments have indicated a role for mechanical force in hybrid domain swing-out and integrin extension [81,85,117,118]. These motions are accompanied by the opening of the highly flexible interface between the β subunit domains I-EGF1 and I-EGF2, and extension of the α-genu in the α subunit. While several different structures of α_X_β_2_ in different crystal lattices show that the β subunit appears to be more flexible than the α subunit [45], *α* subunits contain two regions of high flexibility: the linker between the β–propeller and thigh domain and the knee at the bend between the thigh and the calf-1 domain, which allows extension of this subunit by a hinging at the *α* knee, or “genu” [98]. For the β subunit, the region around the EGF domain is almost plastic, including the linker between EGF1 and EGF2 the PSI/hybrid and hybrid/I-EGF1 junctions [98]. In the β-I/hybrid region, there is evidence of significant conformational changes both in the absence and in the presence of force [47,108,119,120,121,122,123,124,125]. During conformational activation of α_IIB_β_3_, the displacement of the α7-helix in the hybrid domain leads to a ~60° reorientation between the β-I and hybrid domain, and the α and β leg knees become separated by 70° due to the transmission through the firmly linked PSI domain in the upper β leg [108].

One hypothesis for how force induces these conformational changes is that force physically propagates from the ligand binding site and the intracellular β tail across the integrin structure, with sequential stretching and deformations of the interconnections between its rigid domains. In this case, rapid force transmission across the protein leads to rigid-body domain displacements without affecting secondary structures. Force, propagated along the linkers between domains, provides the energy for overcoming the barrier(s) between bent and extended conformations. This concept of rigid body-based structure, in which domains act as rigid units with articulation points located between them, was proposed from high-resolution reconstructions of α_IIB_β_3_ and α_v_β_8_ integrins, and of a β_2_ subunit fragment [126,127,128]. A second hypothesis is that physical force propagation across the structure sequentially stretches and deforms the individual domains, so that the more flexible domains also undergo conformational changes. The presence of flexible domains within integrins has been reported for α_5_β_1_ and α_X_β_2_ integrins [51,129]. These integrins have also been shown to bind ligands in conformations that are different from the extended conformation [90,91]; thus force may alter domains structure in addition to their relative positions. The flexible domains include leg domains, such as the thigh, PSI and EGF domains, that are distant from the ligand binding site and can alter allosteric equilibria during the conformational transition [130]. A third hypothesis is that force does not directly propagate across the entire structure but is absorbed at the site of application. In this case, the fast extension or dynamic rearrangement of a flexible domain buffers this force by directly converting it into local intramolecular adjustments that more slowly drive the global change in conformation of the protein. In support of this picture, steered molecular dynamics simulations of integrin conformational activation have shown that an inward displacement of the MIDAD metal ion is coupled to the movement of the α-7 helix from the up to middle and down positions, and that this movement correlates to different degrees of integrin conformational activation [131]. This finding has also confirmed previous crystallography and biomembrane force probe experiments [132,133]. The main difference between these three possible pathways relies on how exactly force propagates (or does not) from the ECM or cytoskeletal binding sites to allosterically drive the whole structure into a new conformation.

Both closed and open conformations of integrin can bind ligands [125]. Integrin ligand binding may be favored kinetically for the extended-closed conformation, whose ligand-binding site is more accessible to ligands than the bent closed conformations (Figure 2) [125]. With the extended-closed integrin anchored to the cytoskeleton and bound to an extracellular ligand, if force is normal to the plasma membrane, integrin conformation may be shifted to the extended-open conformation (Figure 4). In this case, ECM ligand pulling of the β subunit promotes lateral extension of the hybrid domain away from the ligand binding interface. This extended-open conformation of integrin presents slow ligand unbinding rate under force, following catch bond behavior [79,114].

It is important to note that, although this pathway is generally shared across several β_3_ integrins (e.g., α_v_β_3_ and α_IIB_β_3_), whether or how it applies to other integrins is uncertain, in particular, α-I domain containing integrins such as α_X_β_2_, or α_4_β_1,_ and α_5_β_1_. Truncated integrin α_5_β_1_ ectodomains do not undergo the large conformational change in the presence of the RGD peptide alone [53]. In these cases, the closed conformations of β_1_ integrins bind ligands with higher on-rates and higher ligand binding affinity than the extended-open conformation [90,91]. α_4_β_1_ was shown to be easier to activate, but its high-affinity conformation binds fibronectin ~100- to 1000-fold lower affinity than α_5_β_1_ [41]. Moreover, negative-stain electron microscopy (EM) has hinted that the α_5_β_1_ ectodomain may not reach a fully bent conformation [134]. Thus, for these integrins, the conformational transition from bent closed to extended-open may occur mainly after force is applied. In α-I containing integrins, the high flexibility of the α-I domain itself is tightly coupled with the β-I domain to fine-tune ligand binding affinity. Initial coupling between the β-I and αI domains might stabilize a partially open state of the α-I domain with a closed MIDAS, which on ligand binding converts to the open α-I state with an open MIDAS [45,132]. These observations challenge the view that bent or partially bent integrins present an occluded ligand-binding site and suggest that the receptor response to force varies across families.

It is also hypothesized that force from the actin cytoskeleton is exerted on extended-closed integrin, with the force parallel to the plasma membrane [135,136]. This lateral force then reorients the β helix relative to the α helix to induce the extended-open conformation (Figure 5). In this model, lateral force on the β tail is transmitted through the lower β-leg domains to the hybrid domain, which swings out and assumes an orientation like that in the open headpiece.

## 7. Conclusions and New Areas of Research

To summarize the above discussion, integrin conformational dynamics in which bent, low affinity conformations convert to extended, open, high affinity conformations are modulated by cytoplasmic adapter and ECM ligand binding, and then further modified by tension applied through this linkage. Accelerated conversion to or stabilization of high affinity conformations contributes to cellular mechanotransduction, such as stiffness sensing or responses to applied tension. However, major questions remain about the exact conformational paths that mediate these transitions, how forces are transmitted through the integrin structure, and how they impact these transitions. It is also unclear whether integrins under tension might access conformations that are rare in unloaded conditions.

In the absence of experimental techniques to obtain dynamic, atomic-level insights into the integrin activation pathway particularly under tension, addressing these questions will remain limited to computational modeling based on Xray or cryo-EM structures of the unliganded closed-hinge and the ligand-bound open-hinge β3-integrin headpiece domains [43]. Molecular dynamics (MD) simulations of the β3-integrin headpiece domains have illustrated the Ångstrom-level structural pathway of ligand-induced hinge-angle opening [114]. Incorporating force into these calculations and then experimental verification of predictions is the current state of the art.

Lastly, how integrin conformational dynamics impact cell functions is a major unexplored area. Integrin-based adhesions of platelets and leukocytes are “off” in the vasculature and “on” when these cells are activated. Directional cell migration requires local control of integrin activation and adhesion strengthening at the cell front and deactivation and detachment at the rear [135]. Indeed, adhesion strengthening is reportedly confined to the cell front [3]. Cell adhesions through integrins must also be coordinated with cytoskeletal dynamics to enable integrins to connect to the force-exerting actin fibers and provide traction and propulsive forces for cell migration [39,76,135]. These processes appear to differ for different integrins. For example, α4β1 is reportedly activated by lower forces or at a lower concentration of adaptor proteins compared to α5β1, perhaps in keeping with α4β1 mediating leukocytes adhesion and migration that occurs under low tension [41]. Thus, understanding how force-dependent conformational dynamics underlie adhesion dynamics is an important area for future study. Addressing these questions likely requires new methods in structural and mechanobiology but will be foundational for understanding integrin functions in physiology and disease.

## Figures and Tables

**Figure 2 cells-11-03584-f002:**
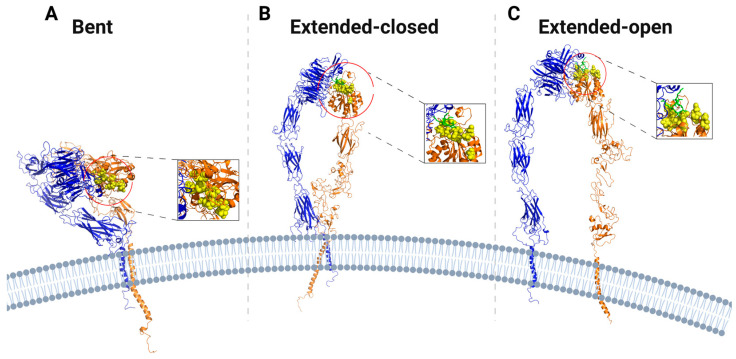
Reorientation of the ligand-binding interface during conformational activation of integrin αIIbβ3. (**A**). Ribbon representation of the platelet αIIbβ3 integrin in the bent conformation, with the ligand-binding interface (yellow, vdW representation) oriented towards the lower legs. (**B**,**C**). Ribbon representation of integrin in intermediate and extended-open conformations, with the ligand-binding interface (yellow, vdW representation) oriented away from the lower legs. The α subunit and the β subunit are represented in blue and orange, respectively. All structures are acquired from cryo-EM reconstructions of αIIbβ3 integrin [24].

**Figure 3 cells-11-03584-f003:**
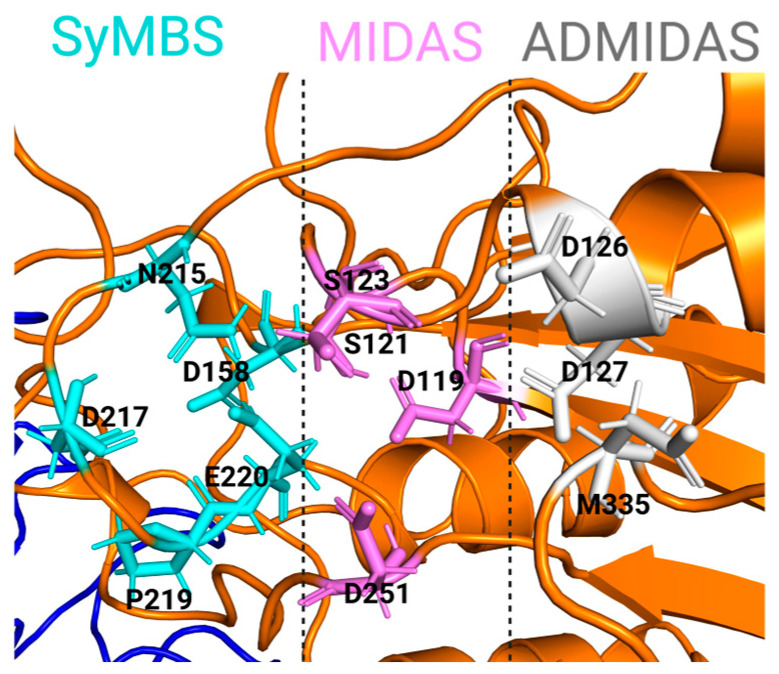
Structure of the metal ion-binding sites of bent αIIbβ3. The metal ion binding sites correspond to the following residues: MIDAS (in pink) includes E 220, S 121, S 123, D 119, D 251; SyMBS (in cyan) includes D 217, N 215, D 158, P 219; ADMIDAS (in grey) includes D 126, D 127, M 335.

**Figure 4 cells-11-03584-f004:**
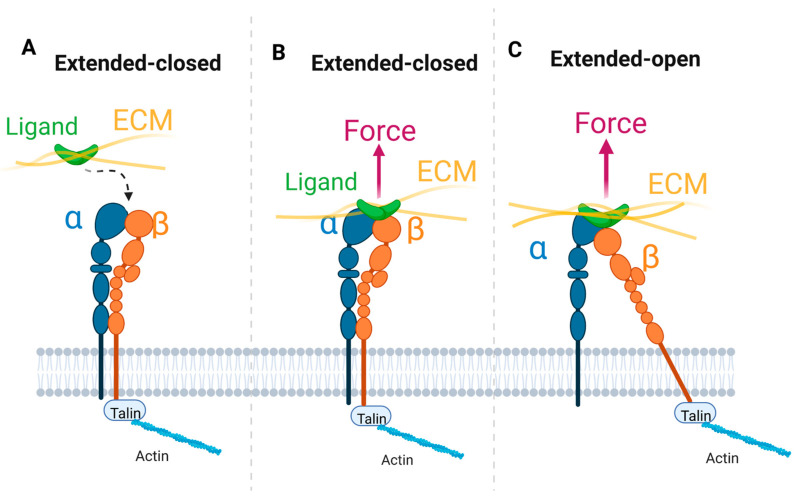
Effect of ECM ligand binding and pulling on integrin conformation. (**A**). Schematics of talin-bound integrin in extended-closed conformation. (**B**). Once the extended-closed conformation of integrin attaches to an ECM ligand, a membrane-normal force is exerted. (**C**). ECM pulling may shift the conformation to the extended-open with swing-out of the hybrid domain. A single integrin is shown here; note, however, that integrins usually function in clusters.

**Figure 5 cells-11-03584-f005:**
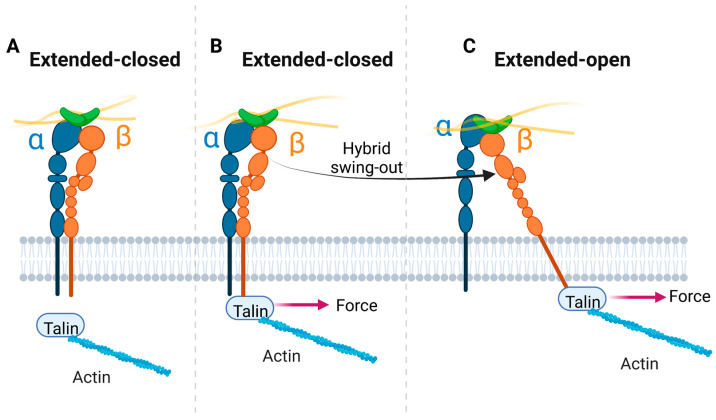
Effect of talin binding and pulling on integrin conformation. (**A**). Schematics of ECM-bound integrin in extended-closed conformation. (**B**). Once the extended-closed conformation of integrin attaches to intracellular talin, a membrane-parallel force is exerted on the β tail. (**C**). The connection of talin to the actin cytoskeleton provides lateral pulling of the β tail, which reorients the transmembrane β helix relative to the α helix, to induce the extended-open conformation. A single integrin is shown here; note, however, that integrins usually function in clusters.

## Data Availability

Not applicable.

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
