# Peer review of "Integrin Conformational Dynamics and Mechanotransduction"

_cells, 2022, doi:10.3390/cells11223584_

Round 1

Reviewer 1 Report

This is comprehensive review detailing the mechanisms and cellular and physiological consequences of integrin-mediated mechanotransduction.

Comment: The use of "integrin" instead of "integrins" in places is awkward - maybe typos. For example see lines 141 -142.

Author Response

Response to Reviewer 1

This is comprehensive review detailing the mechanisms and cellular and physiological consequences of integrin-mediated mechanotransduction.

We thank the reviewer for this positive comment.

Comment: The use of "integrin" instead of "integrins" in places is awkward - maybe typos. For example see lines 141 -142.

We have corrected this expression.

Reviewer 2 Report

The idea of writing a review to summarize and clarify the main aspects in integrin conformational changes during adhesion events is very good.

Anyway the manuscript is very confused, there is no clear differentiation among the integrin classes and in each paragraph, observations on different receptors are mixed together.

Another issue is related to the described papers, that are quite old. Only less than 10% of the references have been published in the last 5 years.

Only paragraphs 6 and 7, describing the effects of force on integrin conformation and the new perspective in this field, seem to present some elements of novelty, even if the references are quite old even for these duscussions. 

The papers reported before 2018 have been extensively reviewed and discussed previously as for instance in the review reported by Zhao et al. (Talyn-1 interaction network in cellular mechanostransduction, Internationa Journal of Molecular Medicine, 49:5, 60, May 2022) or in the other 30 reviews published in the last five years in related fields.

The authors should repeat the bibliographic search and review more recent papers before resubmitting the mianuscript.

Reviewer 3 Report

This is an excellent and very well-written review focused on the conformational regulation of integrin function and mechanotransduction. The authors make a detailed summary about different conformational stages of integrins and the classical model of integrin activation. 

No special edition or corrections are needed, but the authors could consider discussing the following issues in the review:

1) The classical model integrin activation and ligand binding is based on studies related to many different integrins, still it is possible or even probable that there are import differences between integrin subfamilies. For example to dynamic nature of integrin alphaI domains (as show e.g. Xie et al., EMBO J. 29: 666, 2010) may indicate significant functional differences between alphaI domain containing and other integrins (in ligand binding and mechanotransduction).

2) How is integrin function and mechanotransduction affected by shear stress? Many integrins are expressed on platelets and inflammatory cells that are circulating in blood. 

3) The review shortly mentions that integrins in nonactivated conformations may also interact with ligands. The recent paper sheds new light to this idea: Li et al. Elife 2021 Dec 2;10:e73359. Can ligands activate integrins without preactivation through inside-out signalling? 

4) In Figure 4. The model shows only one integrin heterodimer. Normally integrins are in clusters in focal adhesion sites. Furthermore, plasma membrane is a very dynamic structure. Is it possible to predict the effects of these facts on the model? 

Reviewer 4 Report

The authors presented a concise review of the structure-function relationship of Integrin. In this review, the authors have discussed the basic features of integrin conformational states and the impact of mechanical forces on these states. In their conclusion, they have suggested the amount of work and important questions yet to be addressed.

Author Response

We thank the reviewer for the positive evaluation of our work.

Round 2

Reviewer 2 Report

The revised manuscript is now suitable for publication